# Partial Breast Reirradiation for Breast Cancer Recurrences After Repeat Breast-Conserving Surgery with Proton Beam Therapy: The Prospective BREAST Trial (NCT06954623)

**DOI:** 10.3390/jcm14103416

**Published:** 2025-05-13

**Authors:** Eva Meixner, Semi Harrabi, Katharina Seidensaal, Beata Koczur, Thomas Tessonnier, Adriane Lentz-Hommertgen, Line Hoeltgen, Philipp Hoegen-Saßmannshausen, Fabian Weykamp, Jakob Liermann, Juliane Hörner-Rieber, Jürgen Debus

**Affiliations:** 1Department of Radiation Oncology, Heidelberg University Hospital, Im Neuenheimer Feld 400, 69120 Heidelberg, Germany; semi.harrabi@med.uni-heidelberg.de (S.H.); katharina.seidensaal@med.uni-heidelberg.de (K.S.); adriane.lentz-hommertgen@med.uni-heideberg.de (A.L.-H.); line.hoeltgen@med.uni-heidelberg.de (L.H.); philipp.hoegen@med.uni-heidelberg.de (P.H.-S.); fabian.weykamp@med.uni-heidelberg.de (F.W.); jakob.liermann@med.uni-heidelberg.de (J.L.); juliane.hoerner-rieber@med.uni-duesseldorf.de (J.H.-R.); juergen.debus@med.uni-heidelberg.de (J.D.); 2Heidelberg Institute of Radiation Oncology (HIRO), 69120 Heidelberg, Germany; 3National Center for Tumor Diseases (NCT), 69210 Heidelberg, Germany; 4Heidelberg Ion Therapy Center (HIT), Im Neuenheimer Feld 450, 69120 Heidelberg, Germany; beata.koczur@med.uni-heidelberg.de (B.K.); thomas.tessonier@med.uni-heidelberg.de (T.T.); 5German Cancer Research Center (DKFZ), Clinical Cooperation Unit Radiation Oncology, Im Neuenheimer Feld 280, 69120 Heidelberg, Germany; 6Department of Radiation Oncology, University Hospital Düsseldorf, 40225 Düsseldorf, Germany

**Keywords:** reirradiation, repeat irradiation, dose constraints, oncologic outcome

## Abstract

(1) **Background**: The management of ipsilateral breast cancer recurrence depends on the extent of the tumor, and staging results, and mastectomy is currently the standard of care for previously irradiated patients. Studies are increasingly investigating suitable candidates for the repeated use of breast-conserving approaches as an alternative to mastectomy. But this includes the crucial necessity for curative reirradiation (Re-RT). The therapeutic challenge in reirradiation involves finding a balance between tumor control and the risk of severe toxicity from cumulative radiation doses in previously irradiated organs. Re-RT options include the use of brachytherapy, intraoperative radiotherapy, or external beam RT with photons or electrons. The application of particle therapy using proton beam therapy represents an innovative radiotherapeutic technique for breast cancer patients that might offer advantageous physical properties, a superior dose reduction to adjacent organs-at-risk, and effective target volume coverage with lower integral doses to the patient’s whole body. In addition, this technique could potentially offer higher radiobiological effects and tumor responses. (2) **Methods**: The BREAST trial (NCT06954623) will be conducted as a prospective, single-arm, phase II study in 20 patients with histologically proven invasive breast cancer recurrences after repeat breast-conserving surgery and with an indication for local reirradiation. The patients will receive partial-breast re-RT with proton beam therapy in 15 once-daily fractions up to a total dose of 40.05 Gy(RBE), delivered with active raster scanning. The required time interval will be 1 year after previous RT to the ipsilateral breast. (3) **Results**: The following results will be reported: The primary endpoint is defined as the cumulative overall occurrence of (sub)acute skin toxicity of grade ≥ 3 within 6 months after the start of re-RT. Secondary outcome includes an analysis of the local, regional, and distant control, progression-free and overall survival, quality of life, and cosmesis. The explorative and translational objectives of this study include planning comparisons to other RT techniques and irradiation types, dosimetric evaluations, analyses of radiological imaging features, and translational assessments of cardiac toxicity biomarkers and tumor markers. (4) **Conclusions**: Overall, the aim of this study is to evaluate the potential of proton beam therapy for partial breast reirradiation and to establish the underlying data for a randomized trial.

## 1. Introduction

Breast cancer is the most common malignancy in women [1] in the world, with about 70,000 women diagnosed in Germany in 2020 [2]. Breast-conserving surgery (BCS) with adjuvant whole-breast irradiation (WBI) is currently the standard of care in the oncological treatment of primary breast cancer and offers an equivalent alternative to mastectomy. Adjuvant systemic therapy is applied according to the estrogen receptor (ER) and progesterone receptor (PR), as well as the human epidermal growth factor receptor 2 (HER2), statuses and further depending on the tumor’s extent and nodal status [3]. The primary aim of adjuvant radiotherapy (RT) after BCS is to improve the local control and thus improve the overall survival and breast cancer-specific mortality, which has been confirmed in several randomized controlled trials and meta-analyses [4,5]. However, nodal-negative women have a 10-year risk for local recurrences after RT of up to 15.6% [4], with most local recurrences occurring in the proximity of the primary tumor bed [6]. The management of ipsilateral breast cancer recurrence depends on the extent of the tumor, disease, and staging results at the time of recurrence, and mastectomy is currently the standard of care for previously irradiated patients [7].

### 1.1. Mastectomy Versus BCS

Historically, salvage mastectomy has been the only curative treatment approach in patients with ipsilateral breast cancer recurrences in previously irradiated breasts. Studies are increasingly investigating suitable candidates for the repeated use of breast-conserving approaches as an alternative to mastectomy, aimed at avoiding mutilating surgery. With appropriate patient selection, some studies have proven equivalent survival outcomes after repeat BCS [8,9], with the 10-year survival after salvage mastectomy being about 65.7%, compared to that of salvage breast-conserving surgery, which is 58.0% [10]. But this treatment approach includes the crucial necessity for curative reirradiation (re-RT) [11]. The therapeutic challenge in re-RT involves finding a balance between tumor control and the risk of severe toxicity from cumulative radiation doses in previously irradiated organs.

### 1.2. Common re-RT Options and Toxicity: Brachytherapy, Intraoperative (Electron) RT, and External Beam RT

Re-RT options mostly comprise smaller target volumes of partial-breast irradiation (PBI) with the use of brachytherapy (BT) [11,12], intraoperative (electron) radiotherapy (IO(E)RT) [13,14,15,16], or external beam RT (EBRT) [17,18,19,20,21] with photons or electrons. In this context, accelerated concepts further allow for an increase in the dose per fraction combined with a decrease in the overall treatment time. Several different techniques have been used to deliver accelerated partial-breast irradiation (APBI). Brachytherapy was one of the first techniques studied for APBI in re-RT regimens with robust results [22] and can be applied by balloon-based or multicatheter interstitial techniques by the insertion of thin catheters into the former tumor region. But BT involves surgical intervention, antibiotic prophylaxis, and in-ward treatment, as well as significant requirements for its technical feasibility regarding equipment and the expertise of physicians and physicists. Further, some tumor locations near the chest wall or axillary region sometimes limit or prevent the accessibility of the application. The single-dose or electron intraoperative irradiation (IO(E)RT) of the tumor bed is another option for partial-breast RT in the primary situation [23,24,25] and for re-RT [13,14,15,16]. IO(E)RT is applied to the tumor bed after tumor resection during BCS. This ensures the reliable identification of the tumor bed without geographical mismatch due to oncoplastic tissue displacement and replaces long-lasting treatments. Thirdly, the application of percutaneous, external beam RT with a conventional linear accelerator using photons or electrons is universally available and a convenient, non-invasive re-RT approach. However, the dose of the prior radiation exposure of the organs often represents a limiting factor in EBRT, as doses to the surrounding OARs are higher than for re-RT with BT or IORT.

Depending on the time interval since previous RT [26] and the extent of the reirradiated volume, as well as the applied re-RT technique, toxicity rates range largely: For brachytherapy, high-grade (≥3) toxicity rates for re-RT range from about 9% for acute skin [22] to 12% [27]–17% [28] for late fibrosis, and 1% [22]–10% [29] for G4 toxicity. For IO(E)RT, the late grade 3 fibrosis is reported to be about 21% [16], and for EBRT, the grade 3 acute toxicities (skin, esophagitis, wound dehiscence) range from 7% [30] to 24% [31]–35% [32] for EBRT [20,21,32,33], including late breast volume asymmetry, identified in 12% [34], acute and late grade 4 ulceration (1–2%) [30,31], and even grade 5 (treatment related deaths) in 1.2% [31]. Further, grade 3 toxicity with rib fractures (1.2%) [32], pneumonitis (1.2%) [32], lymphedema (1.2%) [32], infection 1.7% [21] are described. Our institutional results of 25 patients treated with reirradiation to the partial-breast, chestwall ± regional nodes using 3D-conformal, intensity-modulated, MR-guided radiotherapy and helical tomotherapy resulted in acute grade 3 and 4 toxicity in 8% and 4%, respectively. Late grade 3 toxicity was present in 12% and significantly associated with a shorter time interval since initial RT and the use of helical tomotherapy. No late grade ≥ 4 toxicity was observed [35].

Cosmetic outcomes were rated as excellent in about 45.5% [36]–64.1% [12] after BT and as good/excellent in about 60% of cases after EBRT [34].

Overall, considering these results, acute grade 3 toxicity from commonly used and widely available techniques ranged largely from 7–35% depending on various factors, such as the time interval and dose from prior RT, the extent of the reirradiated volume, and the application of concomitant chemotherapy or hyperthermia [32], with only a few studies reporting grade 4 or 5 toxicity.

### 1.3. Oncological Outcome After Repeat BCS Plus re-RT

The oncological outcome after repeat BCS followed by postoperative re-RT achieved promising results: the 5-year local recurrences rate was reported to be 5% [21], resulting in 5-year local control and 10-year regional-relapse-free survival of about 93% [28] and 94% [37], respectively.

In various studies, the 5-, 6-, 8-, and 10-year overall survival was stated to be about 87% [28]–95% [12], 91.2% [38], 90% [34], and 94% [37], respectively. Further, the 6- and 10-year metastasis-free survival was 96.4% [38] and 89% [37], respectively. But this approach requires critical patient selection for suitable re-RT candidates in advance.

### 1.4. Patient Selection

Factors that guide adequate patient selection for partial-breast re-RT, besides the patient’s preferences, are the recommendations of the ASTRO [39], GEC-ESTRO [40], and DEGRO [41] societies for the general PBI for primary breast cancer patients, as well as the results of prior re-RT studies and international societies [42], which correlate oncological outcomes with recurrent tumor characteristics. On the basis of this data, the following criteria are commonly considered to indicate suitable candidates for a second wide excision plus re-RT:-Invasive breast cancer recurrences or second primary breast cancer after prior RT, as well as ductal carcinoma in situ with a tumor size of less than 2 cm [12,37,43] or up to 3 cm [34,36,40,44] in precise imaging (at least a standard mammogram) that accurately evaluates the location and size [36].-Tumors with a positive ER status [10,12], any type of hormonal receptor or tumor grade [36], and a positive epidermal growth factor receptor (HER2) status, except in highly proliferated tumors in young patients who are unsuitable to receive anti-HER2 therapy systemic therapy [36].-Clinically node-negative patients at relapse [10,34,36,40] and with non-metastatic disease [36].-A negative resection margin/R0 resection, defined as “no ink on tumor” [26,36,38,40].-A time interval of >12 months between the first and second course of RT [12,34,36].

The inclusion of patients to repeat BCS with re-RT with the following tumor characteristics, that are often described as high-risk APBI group factors [37,38,40], should be considered very critically: Diffuse lobular tumor histology [36,40], the presence of lymphovascular invasion [36,38,40], known germline BRCA1/2 mutation [10,36] requiring mastectomy or inadequate recovery from prior radiotherapy with high toxicity. Overall, there are only limited data available on optimal recommendations and sequencing of concomitant additional treatments of hyperthermia or systemic therapy for ipsilateral breast tumor recurrences [45].

### 1.5. Proton Beam Therapy

The application of particle therapy using proton beam therapy (PBT) represents an innovative radiotherapeutic technique for breast cancer patients [46]. The dosimetric profile of PBT with the Bragg Peak offers advantageous physical properties and has proven to be superior to photon-based techniques in respect of the dose reduction to adjacent organs-at-risk (OARs) and effective target volume coverage with lower integral doses to the patient’s whole body [47]. In addition, this technique could potentially offer higher radiobiological effects and tumor responses [48], which is particularly advantageous in potentially more therapeutically resistant disease biology in breast cancer recurrences.

The prospective data of proton beam therapy in breast cancer are sparse and mostly focus on primary RT after the first resection. Jimenez et al. [49] reported the promising results of a phase II study for 3D-conformal and pencil beam scanning proton beam therapy in 69 patients for primary breast cancer patients. The 5-year locoregional failure and overall survival were 1.5% and 91%, respectively, with low acute skin toxicity grade 3 (3%), subacute grade 3 seroma (1%), infection (1%), grade 2 pneumonitis (1.4%), and grade 1 subacute/late rib fracture (7%). The prospective phase I/II study for primary RT in breast cancer patients by Freedman et al. [44] reported the feasibility of double scattering and pencil-beam scanning proton beam therapy for the application of accelerated partial-breast cancer in 41 patients with only low grades of acute skin (grade 1: 80.5%, grade 2: 7.3%, no grade 3) and long-term toxicity (grade 3: 4.9%). Local recurrence appeared in only one patient. Th cosmesis was rated as excellent in 56%, good in 19%, and fair in 25% after two years, which did not meet the goals for the cosmesis outcome and led to the premature closure of the study.

Overall, the prospective experience related to repeat BCS plus the re-RT of the ipsilateral breast after recurrences is limited [32], and, up to now, most commonly involves the technique of invasive multicatheter brachytherapy.

## 2. Materials and Methods

### 2.1. Study Design

The BREAST (Partial breast reirradiation for breast cancer recurrences after repeat breast-conserving surgery with proton beam therapy, NCT06954623, clinicaltrials, registration date 1 May 2025) trial will be conducted as a prospective, single-arm, phase II study including 20 patients with histologically confirmed recurrent (or new primary) ipsilateral invasive breast cancer or ductal carcinoma in situ after the prior RT of the ipsilateral breast and with an indication for partial-breast re-RT. The patient and tumor characteristics are reported in the inclusion criteria in Table 1. Further, the patient and tumor characteristics, such as the histological subtype, hormonal and Her2neu receptor statuses, Ki67 proliferation index, the presence of lymphangiosis, tumor size, resection margin, patient age, and comorbidities will be assessed and reported. The patients will receive partial-breast re-RT with proton beam therapy, delivered with active raster scanning, in 15 once-daily fractions up to a total dose of 40.05 Gy(RBE) (relative biological effectiveness). The recruitment of the patients will start in 2025. The recruitment of the patients is planned over a time period of 36 months. The overall duration of the trial is expected to be approximately 48 months. The inclusion and exclusion criteria are described in Table 1.

### 2.2. Trial Objectives and Assessment

The specific purpose of this trial is to analyze the toxicity of partial-breast re-RT using proton beam therapy in patients with ipsilateral breast cancer recurrences after repeat breast-conserving therapy, who had received a prior course of RT to the breast. Figure 1 shows a flowchart of the BREAST study.

#### 2.2.1. Primary Objective

An overall cumulative acute/subacute skin toxicity grade of ≥3 CTCAE (Common Terminology Criteria for Adverse Events), NCI (National Cancer Institute), version 5.0, associated with study treatment within 6 months after the start of re-RT.

#### 2.2.2. Secondary Objectives

The treatment response and progression will be defined based on clinical and radiological features, according to the most recent RECIST criteria (RECIST 1.1, Response Evaluation Criteria in Solid Tumors), with the evaluation of local, regional, and distant tumor control, as well as progression-free and overall survival at 1, 2, and 5 years after re-RT. Further, the toxicity (according to the CTCAE criteria) and the quality of life (with the EORTC QLQ-C30 and the supplementary questionnaire module QLQ-BR42 measuring specific quality of life aspects related to breast cancer treatment) will be assessed at baseline and during follow-ups for up to 5 years.

The cosmesis evaluation will include digitizer measurements (BRA, breast retraction assessment using the method by Vrieling et al. [50]) to quantitate the amount of retraction of the treated compared to the untreated breast. Two frontal views of the chest (from the neck to the midabdomen) will be taken, one with hands near the body and the other with the hands raised as far as possible above the head. A profile view of the treated breast (with the arms above the head) is also planned for each assessment. The suprasternal notch, as well as a point 25 cm below in the midline, will be marked (Figure 2).

A quantitative analysis will also be performed by a panel of doctors according to the graphic below. The three components of the indices (breast retraction assessment (BRA), percentage BRA (pBRA), and reference length (Ref)) will be determined as shown below and the resulting values will be calculated:(1)$$BRA=\sqrt{(a_{1}-b_{1}{)}^{2}+(a_{2}-b_{2}{)}^{2}}$$(2)Re⁡f=b12+b22(3)pBRA=BRARef×100

As this digitizer measurement method is reported to potentially miss some possible factors for poor cosmesis outcomes (e.g., if cosmetic changes were mainly due to skin alterations) and is weaker for inferiorly located tumors, another clinical rating system will be applied that assesses the overall subjective satisfaction of the patient and treating doctor in 4 categories: excellent/good/fair/poor overall cosmesis with the assessment of the surgical scar, breast size and shape, nipple position, and the shape of the areola by comparing the treated with the untreated breast.

#### 2.2.3. Explorative and Translational Objectives

The treatment plan and irradiation parameters derived within this study will be used for further analyses, e.g., dosimetric data, cumulative doses, plan adaption frequency, dosimetric comparison to other radiotherapy techniques or irradiation types. Further, the radiological imaging feature extraction of radiologic scans will be used to search for imaging biomarkers for the prediction of clinical outcomes.

Another explorative study goal is to longitudinally assess clinical patient- and tumor-related parameters with blood-based analytics. The translational objectives include an analysis of cardiac toxicity biomarkers and tumor markers. In patients with breast cancer recurrences, which are often intensively pre-treated with cardiotoxic chemotherapies, the laboratory tests within this study will include differential blood counts, Lithium–Heparin-Gel panels (e.g., sodium, potassium, c-reactive protein, Troponin T (TNT), n-terminal pro-brain natriuretic peptide (NT-ProBNP), lactatdehydrogenase, creatinine, glomerular filtration rate, etc.) and tumor markers (carcinoembryonic antigen (CEA), CA15-3 (Cancer Antigen 15-3) [51]).

### 2.3. Treatment Planning and Target Volume Definition

Radiotherapy is administered after a full recovery from surgical resection, usually 4 to 6 weeks after BCS or 6 to 12 weeks after adjuvant chemotherapy. The patients will receive computed tomography (CT)-based treatment planning, with 3 mm slice thickness CTs without a contrast agent. In accordance with institutional standards, patients will be immobilized in a supine and arms-up position using an immobilization WingSTEP (IT V, Innsbruck, Austria) device. For skin scar marking, a radiopaque metallic marker will be used. The CT scanning will be performed with free-breathing and a deep-inspiration breath hold to include the extent of breathing.

For the target definition, a consensus statement and recommendation from the breast cancer working group of DEGRO [41] and GEC ESTRO [52,53] for partial-breast irradiation will be used, considering the following steps and preconditions:-Detailed knowledge of preoperative imaging (e.g., mammogram, MRI, CT scans, etc.), the surgical procedure (including oncoplastic approaches, tumor bed clips, etc.), and the pathological report.-The recommended total safety margin around the surgical bed is 2 cm, considering the size of the surgical resection margins in all directions.-The identification and delineation of the skin scar, surgical clips, and whole visible surgical scar tissue inside the breast.-The identification of the estimated tumor bed, considering and related to the tumor’s localization and size in pre- and postoperative imaging.-The delineation of the CTV (clinical target volume), which is defined as the estimated tumor bed plus the above-mentioned 2 cm margin in the corresponding direction.-The thoracic wall/rib plane and the skin are not part of the CTV.-An additional margin to the CTV will be generated to account for positioning and planning uncertainties of 5–10 mm, depending on individual patient-specific factors or beam angles, resulting in the planning target volume (PTV).-An additional PTV_evaluate (the PTV minus a skin subtraction of 3 mm) for the meaningful analysis of a dose–volume histogram will be generated.

The following OAR structures should be contoured: ipsilateral lung, contralateral lung, heart, left anterior descending artery (LAD), ipsilateral whole breast, contralateral breast, stomach, liver, skin (with a subtraction from the external skin of 3 mm), spinal cord, and the rib plane/chest wall (at least 2–3 cm below and above the PTV).

The radiotherapy will be delivered with a moderately hypofractionated dose of 40.05 Gy(RBE) in once-daily fractions of 5 per week in a single dose of 2.67 Gy(RBE) using active raster scanning proton beam radiotherapy and anterior oblique beams. The proton RBE will be on a fixed RBE relative to high-energy photons of 1.1. The Re-RT in this study will focus on the tumor bed with partial-breast reirradiation.

As the dose is prescribed to the PTV, the PTV_evaluate should receive 95–107% of the prescribed dose, aiming to achieve the most homogenous and conformal distribution and the maximum spare of the OARs (Table 2).

For the verification of the correct patient positioning during proton beam therapy, daily orthogonal X-ray-based image guidance will be performed and a correction applied, if necessary. Further, weekly CT scans will be applied to verify the reproducibility and assess for interval soft tissue swelling or other significant anatomic changes. If a change that affects the dose distribution is noted, an adaptive plan will be generated [46]. This procedure is not study-specific, but it is the usual procedure according to the internal SOP (standard operating procedure).

### 2.4. Statistical Considerations and Sample Size Calculation

The sample size calculation is based on a single-arm study on the primary endpoint, namely the overall occurrence of acute/subacute skin toxicity, CTCAE, of grade ≥ 3, assessed within the first 6 months after the start of the re-RT. The following one-sided null hypothesis is to be tested: $H_{0}:p_{Tox}\geq 25\%$ (i.e., a toxicity rate of 25% for photon re-RT [31] should not be exceeded). For the sample size calculation, a better toxicity rate of 3% [49,56] for proton re-RT was assumed. Based on these assumptions, a required number of 20 patients were calculated for a one-sided binomial test at the level α = 0.025 in order to achieve a power of 1 − β = 0.8, while a one-sided 97.5% confidence interval was calculated. Under these assumptions, the actual level is 0.0243, the actual power is 0.88, and the critical value is 1, i.e., the null hypothesis cannot be rejected if more than one patient experiences a grade ≥ 3 toxicity. The sample size calculation was carried out using the rpact package of the open-source software R (Version 3.0, RPACT; Cologne, Germany).

The populations for analysis include the Intention-to-Treat (ITT) population (all the enrolled patients who fulfilled the in-/exclusion criteria and have submitted their written informed consent and have been treated for at least 1 day) and the complete case population: all the patients in the ITT population who were able to receive the planned therapy in full and whose documentation is correspondingly complete. This is the primary evaluation population for primary and secondary endpoints.

No imputation of the primary endpoint will be conducted. Missing values will be reported descriptively. Limitations associated with this statistical procedure will be reported with emphasis. Analyses of secondary, explorative, and translational endpoints will be descriptive and will include the calculation of appropriate measures reporting the means, standard deviations, medians, 1st and 3rd quartiles, the minimums and maximums for continuous or absolute data, and the relative frequencies for categorical variables. Time-to-event endpoints will be visualized by means of Kaplan–Meier plots and cumulative incidence functions. QoL analyses will be performed according to the corresponding EORTC manual. The safety analysis will comprise a tabulation and summary of (serious) adverse events. All the analyses will be performed using validated software. No interim analysis is planned.

### 2.5. Ethical Aspects

The clinical trial will be carried out in accordance with the current version of the Declaration of Helsinki and has been approved by the local ethics committee of the Medical Faculty of the University of Heidelberg (S-723/2024, approval date: 5 March 2025).

## 3. Discussion

In addition to the treatment of primary breast cancer patients, proton beam therapy has been reported in small, mostly retrospective re-RT studies of patients with breast cancer recurrences with various fractionation schedules and regimens and cumulative doses to various target volumes of the whole breast, the partial breast, the chest wall, regional nodes, or a combination of aforementioned. Being applied as an external beam technique, it offers a non-invasive re-RT option, which appears to be more convenient and comfortable than interstitial surgical BT options and has been listed as a treatment option for patients with an indication for reirradiation by the Particle Therapy Cooperative Group (PTCOG) Breast Cancer Subcommittee [46]. However, recommendations include that clinical trial enrollment should be considered if available (LE: 3, Grade C).

As mentioned before, repeat BCS with adjuvant partial-breast photon re-RT, as performed in the NRG Oncology/RTOG 1014 trial, was reported to result in a low 5-year cumulative breast cancer recurrence in selected patients of 5% [21]. While only a few proton studies with limited follow-ups exist, they assessed the 2-year locoregional-recurrence-free survival in curative patients as being 93.1% [33]–100% [54,55], suggesting a comparable and effective treatment alternative to mastectomy.

Re-RT using proton beam therapy was associated with treatment-related side effects: acute and late skin toxicity grade 3 ranged widely: 4% [32]–7% [21,57]–10% [33]–25% [55]–30% [54]; skin toxicity grade 4: 6.2% [55]; and grade ≥ 3 pain: 13% [57]. Of note, the target volume in these studies was more frequently delivered to larger volumes of the chest wall, including regional nodes, compared to photon treatment, with some studies including hyperthermia treatments. Further, the time interval between the first RT treatment and re-RT was significantly associated with the development of grade 3 toxicity at each time point [33].

Overall, proton beam therapy provided a dosimetric advantage in some studies, but the impact on toxicity and oncological control remains unclear, and the optimal technique, target volume, dose, and fractionation regimen have yet to be defined. Table 3 presents an overview of selected studies (no case reports) on oncological outcomes and toxicity profiles of different currently available reirradiation studies exploring the potential of proton beam therapy in breast cancer recurrences. Of note, there are only limited data addressing the issue of moderately hypofractionated partial-breast proton re-RT, which is the primary target in this study.

In addition to the assessment of the safety and outcomes of proton beam therapy, cardiac toxicity will be evaluated in the BREAST trial. The prevention of cardiac toxicity is essential in postoperative RT for breast cancer patients. Postoperative RT has been shown to increase the mortality due to heart diseases, with a rate ratio of 1.27 [5], and cardiac toxicity in long-term follow-up [61,62]. Radiation-induced damages with the risk of an acute myocardial infarction or an increased rate of diagnoses of coronary artery disease has been described, especially in left-sided tumors [63,64,65]. Modern RT techniques with intensity-modulated photon RT use deep-inspiration breath hold techniques to spare the dose to the heart [65,66]. However, the exact mechanism of damage is not exactly known, and the risk of heart injury is further increased by comorbidities or cardiotoxic systemic agents [67,68]. Optimal RT target coverage to achieve oncological local control needs to be combined with technical improvements for the best sparing of the heart [69]. The assessment of a patient’s individual dose-related risk of heart injury and the early detection of cardiac changes and damages is crucial for therapy, mortality rates, and clinical outcomes. Brain natriuretic peptide levels (BNPs) are cardiac blood markers that are elevated in heart failure, acute coronary syndrome, myocardial infarction, and angina pectoris due to related hemodynamic stress due to dilated ventricles or increased wall tension [70,71]. BNPs and their precursor, pro-B-type natriuretic peptide, are commonly available blood markers, and testing for heart failure is recommended with these biomarkers in international guidelines [72]. A prospective study by Palumbo et al. [73] evaluated the dosimetric effects of the irradiation of left-sided breast cancer patients and found no significant changes in left ventricular ejection fraction (LVEF), while the normalized BNPs increased significantly at 1 and 6 months after the end of RT. This was significantly correlated to the V20, V25, V30, V45, mean dose, and mean heart dose. In another study by D’Errico et al. [74], BNP values one year after RT were significantly increased compared to baseline values, indicating a diagnostic possibility to identify subclinical RT-related toxicity. In another study by D’Errico et al. [75], they compared N-terminal pro BNPs and Troponin plasma levels in patients with postoperative RT in left-sided breast cancer after breast-conserving therapy compared to non-RT matched patients. NT-proBNP was found to be significantly increased after RT and was concluded to be a sensitive biomarker of radiation damage. The results indicated that high doses to a small percentage of cardiac substructures were better related to cardiac toxicity than the mean doses, representing the heart as a serial organ. However, the data on cardiac changes are inconsistent, and some studies [49] have reported no significant decreases for left ventricular ejection fractions or circulating biomarkers. The aim of this study is, therefore, to evaluate diagnostic markers and the potential of proton beam therapy to reduce cardiac toxicity during and after reirradiation. Overall, our study will establish the underlying data for a randomized trial.

## Figures and Tables

**Figure 1 jcm-14-03416-f001:**
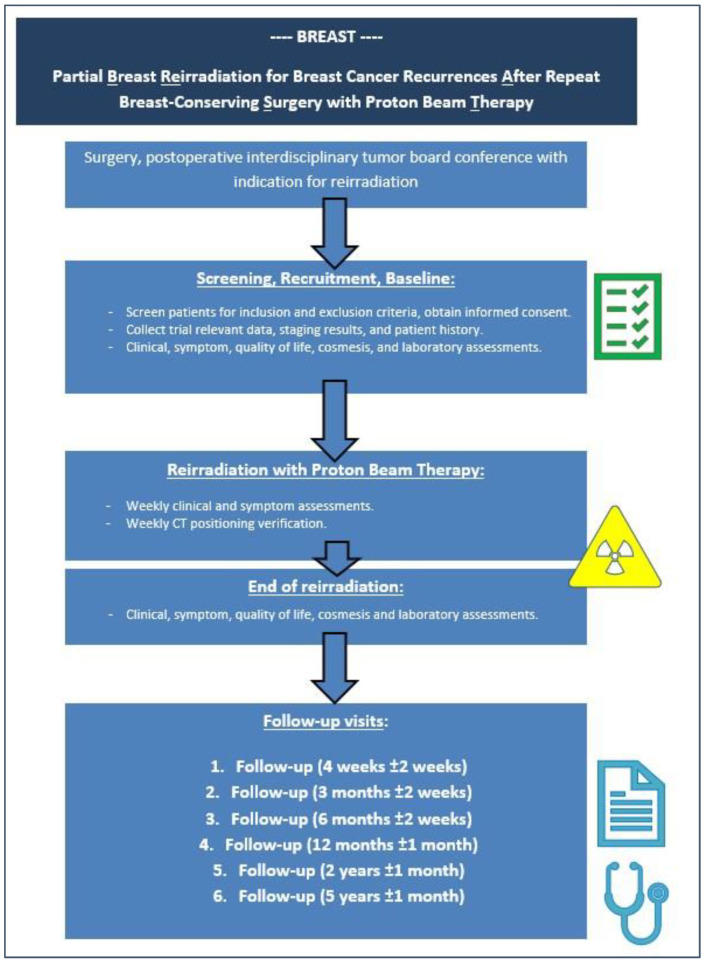
Flowchart of the BREAST trial.

**Figure 2 jcm-14-03416-f002:**
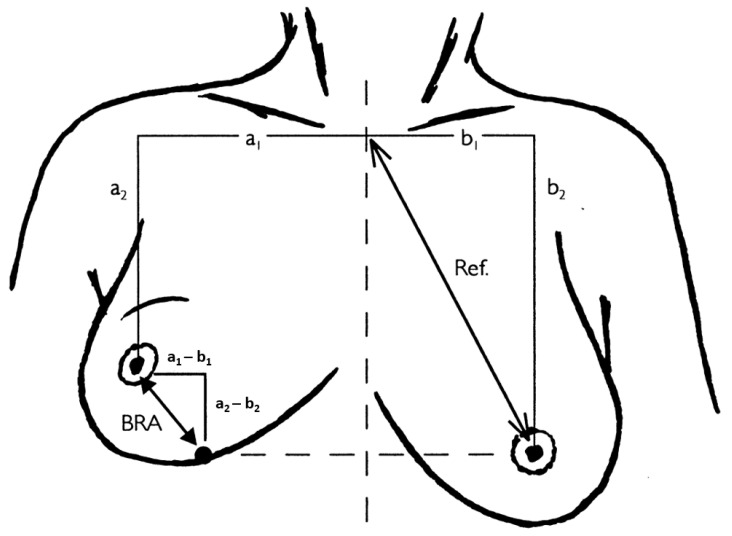
An illustration of the measurements adapted from Vrieling et al. [50].

**Table 1 jcm-14-03416-t001:** Inclusion and exclusion criteria of the BREAST trial.

Inclusion criteria
Histologically confirmed recurrent (or new primary) ipsilateral invasive breast cancer or DCIS after the prior RT of the ipsilateral breast.
Indication for reirradiation after repeat breast-conserving surgery (e.g., lumpectomy, wide excision, etc.).
Tumor size < 3 cm.
Clinically node-negative (cN0).
Negative resection margin (R0).
Time interval: start of re-RT to prior RT ≥ 12 months.
ECOG Performance status ≤ 2.
The ability of the subject to understand the character and individual consequences of the clinical trial.
Written informed consent.
An age of ≥18 years.
**Exclusion criteria**
Distant metastases. Concomitant chemotherapy (concomitant endocrine hormonal therapy is allowed; sequential chemotherapy is allowed). Patients who have not recovered from acute toxicities from prior therapies. Known carcinoma < 5 years ago (excluding a carcinoma in situ of the cervix, basal cell carcinoma, or a squamous cell carcinoma of the skin) requiring immediate treatment interfering with study therapy. Pregnant or lactating women. Participation in another competing clinical study or observation period of competing trials. A history of an active connective tissue disorder (i.e., systemic lupus erythematosus, scleroderma, dermatomyositis, xeroderma pigmentosum, etc.). Medical implants, which are, at the time of reirradiation, not eligible for particle therapy at Heidelberg Ion Beam Therapy Center.

DCIS: ductal carcinoma in situ; ECOG: Eastern Cooperative Oncology Group Performance Score; RT: radiotherapy.

**Table 2 jcm-14-03416-t002:** Recommended dose constraints to OARs (adapted from [33,44,49,54,55]) for reirradiation. Individual dose prescription can differ and will be at the discretion of the treating radiation oncologist.

Organ at Risk/Target	Dose Constraints and Prescription
PTV_evaluate	V95% ≥ 95% (38.05 Gy(RBE))
Heart	Dmean < 2 Gy(RBE), D0.01 cm^3^ < 3 Gy(RBE)
LAD	Dmean < 2 Gy(RBE), D0.03 cm^3^ < 8 Gy(RBE)
Ipsilateral lung	V20 Gy < 20%, Dmean < 10 Gy(RBE)
Skin (3 mm beneath the body surface)	D1 cm^3^~90–95%
Ribs plane/chest wall	D0.01 cm^3^ ≤ 95%
Spinal cord	D0.01 cm^3^ < 5 Gy(RBE)
Contralateral breast, contralateral lung, stomach, liver	ALARA *

* As low as reasonably achievable.

**Table 3 jcm-14-03416-t003:** Selected studies (no case reports) on oncological outcomes and toxicity profiles of different currently available reirradiation studies exploring the potential of proton beam therapy in breast cancer recurrences.

	Author/Citation	Title	Trial Design/ re-RT Target	Time Interval/ Prior RT Dose	Median Dose Proton re-RT/ Cumulative Dose	Toxicity	Oncological Outcome
1	Fattahi et al. [33]	Reirradiation for Locoregional Recurrent Breast Cancer	Retrospective, *n* = 20 PBT (*n* = 52 photons + electrons). Breast/chest wall/nodes/± regional nodes.	6.1 years (73 months). Initial RT: median 60 Gy (range: 50–60.4 Gy).	median 50 Gy(RBE) (1.8–2 Gy) + *n* = 5 additional boost (10 Gy(RBE)) + *n* = 2 integrated boost of 5.6/6.25 Gy(RBE) or hypofractionation 40.05 Gy(RBE) + *n* = 1 additional boost (10 Gy(RBE)) cumulative median EQD2: 103.54 Gy	Overall grade 3 toxicity: 13%. Skin grade 3: 13% acute/3% late.	Median follow-up: 22 months. 2-years LRFS (curative): 93.1%. OS: 76.8%. Distant-metastases-free survival: 59.0%.
2	Choi et al. [54]	Proton reirradiation for recurrent or new primary breast cancer in the setting of prior breast irradiation	Retrospective, *n* = 46. Breast/partial breast/chest wall ± implant/nodes ± regional nodes.	7.1 years (85.5 months) (range: 5–360 months). Initial RT: median 60 Gy (range: 45–66 Gy).	median 50.4 Gy(RBE) (range: 40–66.6 Gy(RBE)) cumulative median 110 Gy(RBE) (96.6–169.4)	Skin grade 2/3: 56.5%/30.4%. Esophagitis grade 2: 8.7%. Late implant contracture 6.7%. Pain grade 3: 7.7%.	Median follow-up: 21 months. No local recurrences. Distant recurrence: 17%. 2-/3-year DMFS: 92.0% and 60.0%. 2-/3-year OS: 93.6%/88.1%.
3	Gabani et al. [55]	Clinical outcomes and toxicity of proton beam radiation therapy for reirradiation of locally recurrent breast cancer	Retrospective, *n* = 16. Chest wall (± implant) ± regional nodes. Concurrent hyperthermia (62.5%).	10.2 years (range: 0.7–20.2). Initial RT: median. 50 Gy (range: 45.0–50.4) + Boost 10 Gy.	median 50.4 Gy(RBE) (range: 41.4–50.4 Gy(RBE)) + *n* = 3 boost 10 Gy(RBE) (range: 10–16 Gy(RBE))	Skin grade 2/3/4: 37.5%/25.0%/6.2%. Fibrosis grade 2/3/4: 12.5%/12.5%/6.2%. Pneumonitis 12.5%. Teleangiectasia 25.0%. Rib fracture 6.2%. Brachial plexopathy 6.2%. Lymphedema 6.2%.	Median follow-up: 18.7 months. No local failures.
4	Sayan et al. [57]	Toxicities and Locoregional Control After External Beam Chest Wall and/or Regional Lymph Node Reirradiation for Recurrent Breast Cancer	Retrospective, *n* = 12 (*n* = 15, but only 80% PBT). Breast/chest wall ± regional nodes.	5.7 years (68.3 months) (range: 7.8–245 months). Initial RT: median 50 Gy (range: 33.5–50.4 Gy).	median 45 Gy(RBE) (range: 42.3–50.4 Gy(RBE))	Skin grade 2/3: 80%/7%. Pain grade 2/3: 20%/13%. Lymphedema 13%. Fatigue grade 2: 40%. Brachial plexopathy: 0%.	Median follow-up: 14 months (range: 1.0–90.5). Locoregional recurrence: 13%. Distant metastases 33%.
5.	Thorpe et al. [56]	Proton beam therapy reirradiation for breast cancer: Multi-institutional prospective PCG registry analysis	Prospective registry, *n* = 50. Breast/chest wall ± regional nodes.	8.7 years (103.8 months) (range: 5.5–430.8 months). Initial RT: median 60 Gy (range: 10–96.7) (only *n* = 43 (86%) had prior RT indication for breast cancer).	median 55.1 Gy(RBE) (45.1–76.3 Gy(RBE)) cumulative median 110.6 Gy(RBE) (range: 70.6–156.8 Gy(RBE))	Overall grade 3: 16%. Acute/late pain grade 3: 10%/4%. Acute/late skin grade 3: 2%/4%. Acute lymphedema grade 3: 2%. Late wound infection: 2%.	Median follow-up: 12.7 months. 1-year LRFS: 93%. 1-year OS: 97%.
6.	LaRiviere et al. [58]	Proton Reirradiation for Locoregionally Recurrent Breast Cancer	Retrospective, *n* = 27. Whole breast/chest wall ± regional nodes.	9.7 years (range: 0.9–37.6). Initial RT: median 46.8 Gy (20–61 Gy).	51 Gy(RBE) (1.5 Gy twice daily) proton double scattering/pencil beam scanning	Acute skin grade 3: 7.4%. Acute pain grade 3: 7.4%. Late skin grade 3/4: 3.7%/3.7%. Late grade 2 rip fractures: 22.2%. Late grade 2 brachial plexopathy: 3.7%.	Median follow-up: 16.6 months. 1-year LRFS: 78.5%. 1-year OS: 78.5%.
7.	McGee et al. [59]	Postmastectomy Chest Wall Reirradiation With Proton Therapy for Breast Cancer	Abstract only, *n* = 22. Chest wall + regional nodes.	12 years (range: 3–36 years). Initial RT: median 60 Gy (10–70 Gy).	median 50.51 Gy(RBE) (range: 45.1–76.3 Gy(RBE)) uniform scanning (*n* = 19) or pencil beam scanning (*n* = 3)	Acute skin grade 2/3: 68.1%/9.1%. Acute grade 2 esophagitis: 31.8%. Acute grade 2 lymphedema: 9.1%. Acute grade 2/3 pain: 9.1%/4.5%. Grade 3 pneumonitis: 4.5%. Rip fracture: 13.6%. Fibrosis grade 3: 9.1%.	Median follow-up: 15 months. 0% local recurrences. 17 months: 1/22 patients with distant metastases.
8.	Choi et al. [60]	Comparative Evaluation of Proton Therapy and Volumetric Modulated Arc Therapy for Brachial Plexus Sparing in the Comprehensive Reirradiation of High-Risk Recurrent Breast Cancer	Retrospective, *n* = 10. Chest wall ± regional nodes.	48 months (range: 12–276). Initial RT: median 50.4 Gy (range, 42.6–60.0).	50.4 Gy(RBE) (45.0–64.4), pencil beam scanning cumulative 102.4 Gy(RBE) (range: 95.0–120.0 Gy(RBE))	brachial plexopathy grade 1: 20%.	Median follow-up: 15 months: 70% alive. 18 months: 2 local recurrences.

DMFS: distant-metastases-free survival. Fx: fractions. Gy: gray. LRFS: locoregional free survival. RBE: relative biological effectiveness.

## Data Availability

The data presented in this study will be available on reasonable request from the corresponding author.

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
