# Peer review of "Partial Breast Reirradiation for Breast Cancer Recurrences After Repeat Breast-Conserving Surgery with Proton Beam Therapy: The Prospective BREAST Trial (NCT06954623)"

_jcm, 2025, doi:10.3390/jcm14103416_

Round 1
Reviewer 1 Report
Comments and Suggestions for Authors
THe authors descriped a new Trial that they will start in 2025.
They want to treat the Re-Rt on the Breast with proton beam to evaluete if they can reduce the Skin toxities.
The trial is well design but now there is olny the general information. The trial is a single center trial and they comunicate his intension.
It is interesting to know that they start but no more. (there are no retrospective planning studies that can compare "in-vitro" in the dose to the skin is really less)
We have to wait for the results (and knwo some techincal information: beam arrangement, beam on time, total time, how they manage the breathing,etc).
It is well written and clear
Author Response
THe authors descriped a new Trial that they will start in 2025.
They want to treat the Re-Rt on the Breast with proton beam to evaluete if they can reduce the Skin toxities.
The trial is well design but now there is olny the general information. The trial is a single center trial and they comunicate his intension.
It is interesting to know that they start but no more. (there are no retrospective planning studies that can compare "in-vitro" in the dose to the skin is really less)
We have to wait for the results (and knwo some techincal information: beam arrangement, beam on time, total time, how they manage the breathing,etc).
It is well written and clear
>>> Thank you very much for reviewing and your comments. As recommended, we will report technical details such as beam arrangement, number of beams, challenges of breathing and positioning, need for re-positioning planning CTs, … with the results.
Reviewer 2 Report
Comments and Suggestions for Authors
This prospective phase II trial investigates partial breast re-irradiation using proton beam therapy after repeat breast-conserving surgery for ipsilateral breast cancer recurrence, aiming to assess safety, toxicity, and tumor control, while exploring clinical outcomes, quality of life, and biomarkers. This study addresses an important clinical challenge with an innovative and well-rationalized approach.
- Can the test of 20 subjects produce statistical significance? The authors should set up a group treated with other radiotherapy methods as a control
- Methods are not detailed enough to allow assessment. Specifically, patient cohort information is totally missing. It is recommended that the authors present the clinical information and tumor characteristics of the 20 screened patients in a table format to improve the evaluability of this prospective trial study.
- The authors should specify and quantify the criteria for cosmetic assessment, such as using the Breast Cancer Treatment Outcomes Scale (BCTOS) and the Harvard scale, and objective measurements using tools like the BCCT
Author Response
This prospective phase II trial investigates partial breast re-irradiation using proton beam therapy after repeat breast-conserving surgery for ipsilateral breast cancer recurrence, aiming to assess safety, toxicity, and tumor control, while exploring clinical outcomes, quality of life, and biomarkers. This study addresses an important clinical challenge with an innovative and well-rationalized approach.
>>> Thank you very much for reviewing and your comments. Please find below a point-by-point list to your comments.
- Can the test of 20 subjects produce statistical significance? The authors should set up a group treated with other radiotherapy methods as a control
>>> The sample size calculation is based for this single-arm phase II study on the primary endpoint overall occurrence of acute / subacute skin toxicity CTCAE grade ≥ 3 assessed within the first 6 months after the start of re-RT. Sample size calculation was carried out using the rpact package of the open-source software R. The following one-sided null hypothesis was to be tested: H0:p_Tox≥25%, i.e. a toxicity rate of 25% for photon re-RT should not be exceeded. For the sample size calculation, a better toxicity rate of 3% for proton re-RT according to current literature (as cited in the protocol) was assumed. Based on these assumptions, a required number of 20 patients were calculated for a one-sided binomial test at the level α=0.025 in order to achieve a power of 1-β = 0.8. Under these assumptions the actual niveau is 0.0243, the actual power is 0.88 and the critical value is 1, i.e. the null hypothesis cannot be rejected if more than 1 patient experiences a grade ≥ 3 toxicity. As prospective and retrospective data is lacking, this data will establish the underlying data and will serve as a starting point for a randomized controlled trial comparing different RT techniques to proton beam radiotherapy.
- Methods are not detailed enough to allow assessment. Specifically, patient cohort information is totally missing. It is recommended that the authors present the clinical information and tumor characteristics of the 20 screened patients in a table format to improve the evaluability of this prospective trial study.
>>> The patient cohort and tumor characteristics are reported in the inclusion criteria in Table 1. To avoid duplicate listings, we have only included some of the patient information in the methods and added some tumor and patient characteristics that will be assessed and reported in the results. All changes in the text are highlighted in yellow.
- The authors should specify and quantify the criteria for cosmetic assessment, such as using the Breast Cancer Treatment Outcomes Scale (BCTOS) and the Harvard scale, and objective measurements using tools like the BCCT
>>> For objective measurement of the cosmesis outcome, we will use photographic cosmesis assessments taken at baseline, at the end of RT and during follow-up visits. Evaluation of breast cosmesis will be performed according to the proposed method by Vrieling et al. to quantitate the amount of retraction of the treated compared to the untreated breast.
Two frontal views of the chest (from neck to midabdomen) will be taken, one with hands near the body and the other with hands raised as far as possible above the head. A profile view of the treated breast (arms above the head) is also planned for each assessment. The suprasternal notch as well as a point 25 cm below in the midline will be marked (Figure 2). Quantitative analysis is performed by a panel of doctors according to the graphic below. The three components of the indices (breast retraction assessment (BRA), percentage BRA (pBRA), reference length (Ref)) are determined as shown below and the resulting values are calculated as added to the protocol.
As this digitizer measurement method is reported to potentially miss some possible factors for poor cosmesis outcome (e.g. if cosmetic changes were mainly due to skin alterations) and is weaker for inferiorly located tumors, another clinical rating system will be applied that asses the overall subjective satisfaction of the patient and treating doctor in 4 categories: excellent / good / fair / poor overall cosmesis with assessment of surgical scar, breast size and shape, nipple position and shape of areola by comparing the treated with the untreated breast.
We added this information and figure 2 to the methods section. Moreover, the supplementary QLQ-BR42 questionnaire module will be assessed.